# Integrated Diabetes Self-Management (IDSM) mobile application to improve self-management and glycemic control among patients with Type 2 Diabetes Mellitus (T2DM) in Indonesia: A mixed methods study protocol

Dewi Murdiyanti Prihatin Putri[1,2☯], Yoyo Suhoyo[3], Ariani Arista Putri Pertiwi[4], Christantie Effendy[5☯]*

1 Doctoral Program of Medical and Health Sciences, Faculty of Medicine, Public Health and Nursing Universitas Gadjah Mada, Yogyakarta, Indonesia, 2 Associate Degree in Nursing Program, YKY Nursing Academy, Yogyakarta, Indonesia, 3 Department of Medical Education and Bioethics, Faculty of Medicine, Public Health and Nursing, Universitas Gadjah Mada, Yogyakarta, Indonesia, 4 Department of Basic and Emergency Nursing Faculty of Medicine, Public Health and Nursing, Universitas Gadjah Mada, Yogyakarta, Indonesia, 5 Department of Medical-Surgical Nursing, Faculty of Medicine, Public Health and Nursing, Universitas Gadjah Mada, Yogyakarta, Indonesia

☯ These authors contributed equally to this work.
* christantie@ugm.ac.id

**Funding:** The Science and Technology Postgraduate Education and Research

## Abstract

The number of patients with diabetes in Indonesia reach 19,47 million in 2021, mostly is caused by the adoption of sedentary and unhealthy lifestyles. Continuous self-management is important in diabetes care. It requires optimal coordination and communication between patients, families, and health care provider. The use of communication technology could be solution to the problem. This study aims to initiate an android-based mobile apps technology as a tool for patient, family, and healthcare provider to optimize patient with T2DM treatment. This study will be conducted in Public Health Centers (PHCs) in Yogyakarta using an exploratory sequential mixed-methods design which is divided into three phases. The first phase will use qualitative descriptive methods. Patients with T2DM, families, nurses, physicians and Health Social Security Agency officers in Indonesia will be involved in a focus group discussion (FGD) and in-depth interviews to understand their needs in optimizing the treatment. The second phase will be the development of an android-based application on the first phase results. The apps will be usability tested by involving experts (heuristic evaluation) and users (think aloud method) to ensure that the apps really meet targeted user's need. In the third phase, we will collect feedback from user after using this apps for three months. The effectiveness of the apps will be measured by self-management improvement and glycemic control of patients with T2DM. The non-equivalent control group design will be applied using a pre-repeated post-test control group. The result of this study will be an Android-based Application which will be called Integrated Diabetes Self-Management (IDSM) app to optimize the implementation of diabetes self-management which can improve

Development Office, Office of the Higher Education Commission provided funding with the grant number is B/67/D.D3/KD.02.00/2019. The grant recipient is Dewi Murdiyanti Prihatin Putri. The funders had no role in study design, data collection and analysis, decision to publish, or preparation of the manuscript. The author does not receive salary from the funder.

**Competing interests:** The authors have declared that no competing interests exist.

glycemic control of patients with T2DM as one of the indicators of the Indonesian Chronic Disease Management Program at PHCs.

## Introduction

### Background

The number of patients with diabetes in Indonesia reach 19,47 million in 2021 [1]. Type 2 Diabetes Mellitus (T2DM) is a metabolic disorder in which insulin secretion is impaired by pancreatic cells and the inability of insulin-sensitive tissues to respond to insulin [2]. The International Diabetes Federation (IDF) revealed that in 2019 the number of patients with diabetes was 463 million adults aged between 20 and 79 years and is likely to increase to 700 million by 2045 [3]. The number of patients with T2DM in Indonesia in 2013 was around 13% of the total population, and continues to increase from year to year [4].

The increasing number of patients with T2DM in Indonesia is caused by the sedentary and unhealthy lifestyles. Poorly managed T2DM has the potential to cause serious complications, like stroke, heart disease, kidney failure or chronic kidney disease (CKD), neuropathies such as blurred vision and other comorbidities, including diabetic ulcers [5].

Diabetes does not only demand immediate medical treatment but also continuous lifestyle changes that require patients to adapt both physically and psychologically in a long term, even for their entire lifetime. Patients with diabetes require holistic and integrative treatment as well as guidance and assistant to be able to manage them selves (good self-management) [6]. Diabetes self-management is an active behaviour in self-care to manage their disease. Patients with chronic diseases are responsible for the daily care of their illness during the time of illness and even for life [6, 7]. The goal of self-management is to normalize blood glucose levels and reduce the risk of long-term complications. Good glycemic control indicates successful short-term self-management [8].

Success in self-management is influenced by the compliance of patients with diabetes in managing their own disease. Adherence involves the patient's voluntary and active involvement in the management of their disease. Compliance behaviour in patients with diabetes includes monitoring blood glucose at home, regulating daily food intake at home or in diet planning, managing home medications, performing recommended physical activities, and foot care. Compliance in the management of diabetes will have an impact on blood glucose levels stability within normal limits [9]. One study showed that the increase in HbA1c in patients was significantly associated with low adherence, low self-care knowledge and lack of contact with health care providers [10]. Non-compliance among patients with diabetes is more often due to lack of information, low awareness about diabetes management, few demands for daily action, emotional stress, low self-commitment and insufficient social support [6, 11, 12].

Social support in diabetes care can be provided by families and health workers. Continuous self-management is very important in diabetes care and requires optimal coordination and communication between patients, families, and nurses [13]. Family involvement to encourage patients with diabetes to adhere to medication, and to behave in a healthy way or modify their behaviour toward a healthier lifestyle is the key to success to control the disease [14]. Family has important roles in increasing patient's independence and quality of life as well as in adapting back to society and regular social life in Indonesia [15]. The family cultural ethos that prevails in Indonesia is collectivist where family members are usually actively involved in caring for other family members who are sick [16, 17]. Self-management also requires proper support

from nurses specially nurses that can enable patients to manage their diabetes confidently and competently. Nurses can facilitate and help patients to set goals and aid in problem-solving for diabetes management [18].

Low monitoring from families and nurses during the COVID-19 pandemic has caused metabolic regulation and control to be disrupted or inadequate so that hypertension and dyslipidemia often occur in patients with diabetes [19]. The existence of a universal protocol regarding social restrictions to minimize the spread of COVID-19 causes the health interventions provided to be considered more effective by using telehealth counselling or with a Smartphone application. This provides additional challenges for nurses to make an innovation by utilizing information technology (IT) in providing health services to patients with diabetes. The involvement of families and nurses through IT is very useful to help improve the self-management of patients with diabetes [20]. The novelty of this study is the development of an android-based application called ISDM which contains the pillars of T2DM management in one application which will connect 3 elements, in T2DM management patients with T2DM, families and nurses. The ability of this app connecting the 3 elements in one mobile application of T2DM management is the integration that never establish before in Indonesia.

### Study aims

The main aim of this study is to determine the effectiveness of an android-based IDSM application in self-management and glycemic control of patients with T2DM. The study's specific aims are: 1) to explore perceptions about understanding, skills, barriers and needs of patients with T2DM, their families and nurses in using android applications to carry out diabetes self-management; 2) to develop an android-based IDSM application that can be used by patients with T2DM, their families and nurses; and 3) to determine the effectiveness of the application in self-management and glycemic control of patients with T2DM.

**Aims of the study protocol.**   This protocol describes a mixed-method study using both qualitative and quantitative methods. This study aims to make an original contribution regarding integrated diabetes self-management by utilizing IT through an android application in improving self-management and glycemic control of patients with T2DM. Publication of the study protocol is the first step in the research process. The authors feel confident that this study protocol can be a reference for other researchers to get an overview of this study.

## Materials and methods

### Study design

This study will use an exploratory sequential mixed-methods design with three phases. The overview of this design is shown in the following Fig 1.

This study is divided into 3 phases: qualitative, qualitative and quantitative methods. This study protocol has been approved by the Medical and Health Research Ethics Committee of the Faculty of Medicine, Public Health, and Nursing, Universitas Gadjah Mada with the number: KE/FK/1068/EC/2021 on September 27, 2021.

### Setting

This study will be conducted at Public Health Centers (PHCs) in Yogyakarta city. Yogyakarta city is one of the 5 districts in Yogyakarta Special Region Province. It has an area of 32.5 km2 (1,02% of the total Yogyakarta province area) and a population of 416,041 people in 2019. It is divided into 14 sub-districts. Every sub-district has 1 to 2 PHCs with a total number are 18 PHCs in Yogyakarta city [21]. Each PHC has a program to engage patients with chronical

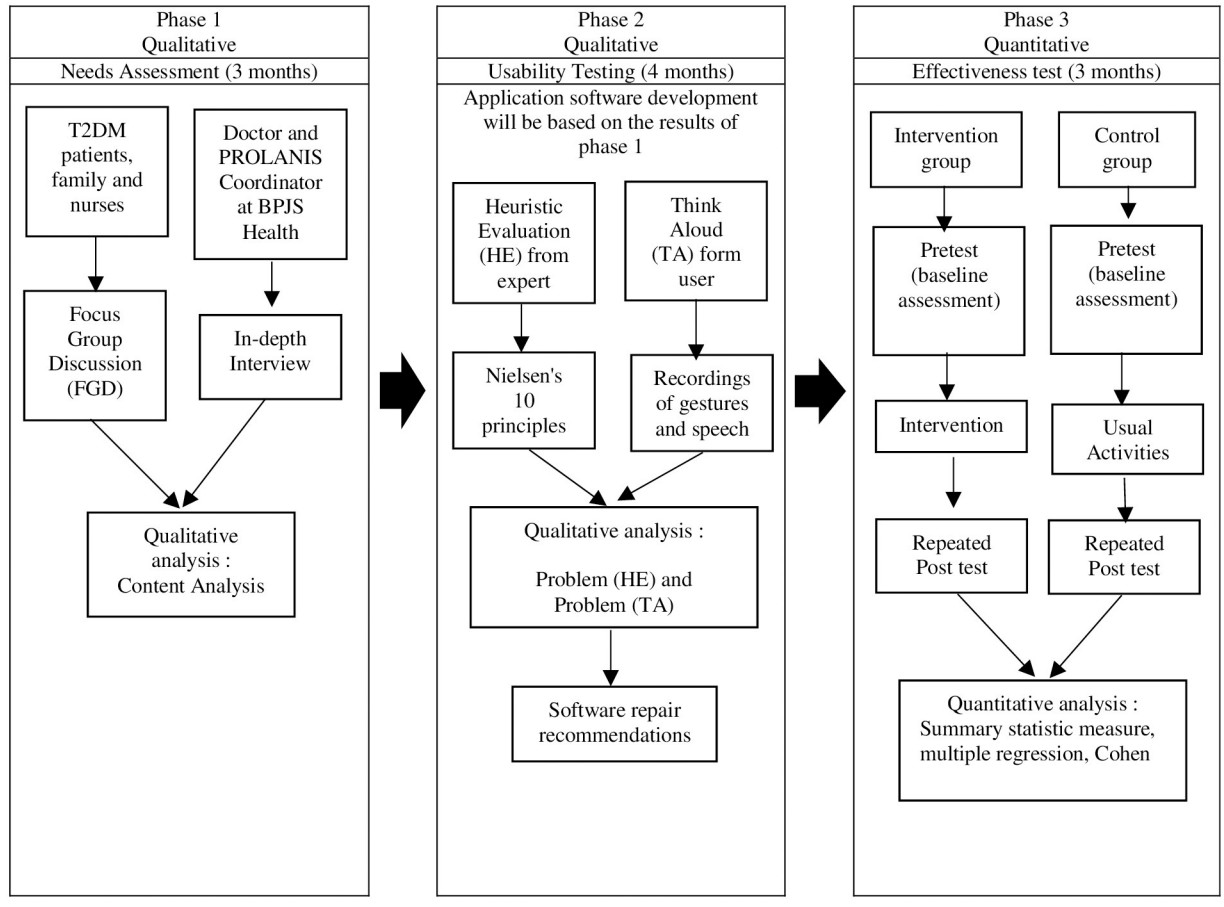

**Fig 1. The exploratory sequential mixed methods design.**

diseases in the disease treatment. The program is called program pengelolaan penyakit kronis shortened to "PROLANIS" which has 58–165 members with the same characteristics in all PHCs. PROLANIS could roughly translated as chronical disease management program.

## Study phases and data collection

The researcher will provide information both verbally and in writing about the purpose, benefits and possible side effects, confidentiality of information, consequence, freedom to refuse if they are not willing and can resign at any time during the study before the research get started. Finally, if participants are willing, then participants are asked to sign the consent form. Each participant has the right to refuse or discontinue participation at any time before or during data collection, without giving reasons.

Phase 1 (Qualitative study). The Focus Group Discussions (FGD) and in-depth interviews will be conducted for data collection. Participants will be divided into three groups, patients with T2DM, families and nurses group. Ten PHCs will be randomly selected, and participants will be selected by the person in charge in PROLANIS at each PHC according to the inclusion criteria. FGDs will be conducted to determine knowledge, abilities, barriers and needs of Patients with T2DM, families and nurses in carrying out diabetes self-management referring to the pillars of DM management. Specifically this phase will investigate targeted user's need in using a mobile application. Patients with T2DM and families will be reached by staffs visiting

their homes and being followed up by phonecall. The FGD will be held for approximately 60 minutes online using zoom video conference platform. It will be guided by researcher and assisted by two nurses with master's degree qualification who act as note taker and observer. Gender, religion, and socioeconomic status are highly considered when we form the FGD groups to ensure that participants will feel comfortable expressing their unique experiences during the FGD process. We will triangulate the data collection by reviewing evidences from other sources to get accurate data. In-depth interviews will be conducted with 10 physicians who treats patients with T2DM at the PHC in Yogyakarta and are in charge at PROLANIS and 2 staffs in charge at the Health Social Security Agency namely [Badan Penyelenggara Jaminan Sosial Kesehatan (BPJS Health)] until data saturation is reached. The interview will be conducted by the researcher for 45–60 minutes in a closed room at PHC. Participant recruitment is determined by the person in charge of PROLANIS from the PHC. Participants will be given an explanation about the study and asked to complete informed consent forms for approval and willingness to participate in the study before the FGD and interviews are started. The FGD and interview guidelines were made based on references to self-management and the pillars of DM management. The FGD will be recorded using zoom recorder and interviews will be recorded using an audio recorder. FGD responses and interviews will be compiled in verbatim transcripts. Peer debriefing and member checking will be done to increase the trustworthiness of the data. We will ensure credibility by doing triangulation of methods (conducting FGD and interviews) and data sources (involving data from patients, families, nurses and physicians. To ensure transferability in this study we will later provide detail, clear, and systematic description of the research results.

Phase 2 (Qualitative study). Phase 2 is focusing on the apps testing. A Heuristic Evaluation (HE) technique will be conducted to ensure that the user interface and user experience of the apps are meeting all heuristic principles. HE is a usability testing that is needed before the application is used. Three experts who have experience in the field of computer science or digital health and understand the heuristic principles will conduct the evaluation in two rounds will be involved in two rounds [22, 23]. In the first round, each expert will conduct the evaluation independently. They will be asked to use the apps and note any violation on heuristic principles they found and rate each violation by 1 (cosmetic problem) to 4 (catastrophe). The apps' module will be provided as wellas a Heuristic principles checklist. In the second round, after the evaluation is complete, a meeting will be arranged to discuss the identified heuristic problems and any disagreement among the experts. The result of the second round will be consensus on the heuristic problem and the rates.

The second technique is a Think Aloud (TA) method will be used to find any problem in the IDSM App. Participants in this phase are Patients with T2DM, families and nurses who were involved in the phase 1. Five participants in each group will be randomly selected [24]. They will be given an explanation and a consent form before the trial started. The TA method is based on the participants' treatment-related responses as indicated by body posture and speech when evaluating with an integrated diabetes management application. Participants will be given a task in accordance with their role. Participants will be asked to say what they are doing, seeing, and thinking while they are using the apps to complete the given task. All the behaviour and words said by the participants will be audio visually recorded. Including the screen of the app. The recording will be compiled into a verbatim transcript. The results of the transcript will be analysed to find problems in the operation of the application software carried out by users [25].

Phase 3 (Quantitative study). This phase will use a non-equivalent control group design, which applies a pre-repeated post test control group design. Participants in this phase are Patients with T2DM, family caregivers, and nurses according to the inclusion criteria. The

experiment and control group are determined by the person in charge of PROLANIS at PHC based on geographic location and PROLANIS activity evaluation data. Data collection will be conducted at two PHCs which has the lowest patients with T2DM number of visits based on data from BPJS Health. However, if the number is not sufficient, it will expand to other PHC. The experiment group will be given IDSM Application, and the control group will use the whatsapp group to communicate or consult directly with a health provider. Self-management will be carried out independently by the patient in his or her home. Monitoring by a health provider will be carried out using whatsapp group and when patient come to the PHC for routine check up. Determination of research subjects will be done by consecutive sampling on patients with T2DM. The sample size was calculated based on the formula for calculating sample size from Lemeshow by considering the hypothesis test for two populations means "two sided" test, with $\alpha$ = 5%; Z(1-$\alpha$) = 1.96; $\beta$ = 20%; Z(1-$\beta$) = 0,84, standard deviation = 0,9 with mean difference = 0.6 [26], then the calculation of the sample size is obtained n1 = n2 = 35 participants. To anticipate the possibility of participants dropping out, a number of samples will be added to keep them fulfilled by the formula: n' = n/(1-f) [27]. As a result, four participants will be added so the number of samples needed is n1 = n2 = 39 participants. The sample will be divided into two groups, each group consisting of 39 patients with T2DM included in the experiment group and 39 Patients with T2DM included in the control group. The sample will be selected according to the inclusion criteria.

After participants are given an explanation beforehand, they will complete an informed consent forms, then they will provide sociodemographic data including religion, marital status, latest education, occupation, and income. The experiment group will first install the app, they will also get IDSM e-module and training how to use the apps before they actually using it. Meanwhile, the control group will only get e-modules. Measurement of diabetes self-management and HbA1C will be conducted before and after the intervention. The Indonesian Version- Diabetes Self-Management Instrument (IDN-DSMI 35) will be used to measure self-management of patients with T2DM. This instrument is valid and reliable with Cronbach's Alpha values of 0.96 and 0.84–0.93 for the subscales. IDN-DSMI can measure self-management behaviour among Indonesian patients with T2DM at PHC [28]. Measurement of self-management scores will be done once a month for three months after the intervention both in the experiment group and in the control group. Based on Lally Gardner (2013), it takes 21 days to form a habit in changing behaviour [29]. The measurement of HbA1C will be conducted before and in the third month after the intervention both in the experiment group and in the control group. Two research assistants will be involved in measuring the self-management score of patients with T2DM. They will previously be given training to ensure the same understanding and perceptions among the to measure self-management and HbA1C. Kappa testing will be conducted to measure inter-rater reliability. The research assistants involved will be persons with minimum education of a bachelor's degree in nursing. The flow of data collection in phase 3 is shown in the following Fig 2.

## Study population

**Inclusion, exclusion, and withdrawal criteria of participants.** Phase 1 (Qualitative study). Participants will be determined using maximum variation purposive sampling technique with the following inclusion and exclusion criteria: 1) Patients with T2DM who is willing to be participant, member of PROLANIS, aged 21 years old or older [30], diagnosed with T2DM for at least 1 year, has a smartphone and can operate it. Exclusion criteria are: patients with Type 1 Diabetes Mellitus (T1DM) or participant who have history of diabetes since childhood, Patients with T2DM who are pregnant, who have comorbidities with Chronic Kidney

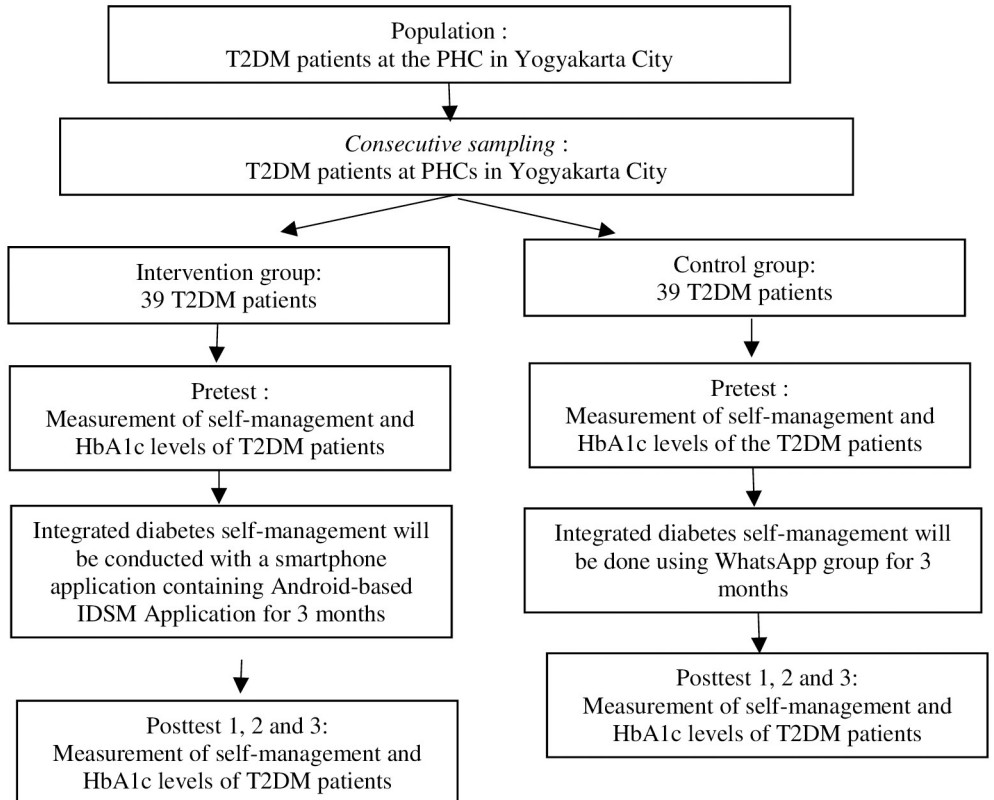

**Fig 2. The quasi-experiment data collection flow.**

Disease (CKD), or mental disorders; 2) Family caregivers inclusion criteria are: members of family aged 18 years old and older, family members who live in the same house as the patient, have a smartphone and can operate it. Exclusion criteria are: family members who work outside the home full day, family members with DM, family members who have mental disorders or who are sick; 3) Nurses: a nurse who carry out the PROLANIS program at PHC where they are on duty, has a smartphone and can operate it. Exclusion criteria are: nurses who are on leave from work; 4) Physician: a physician who treats patients with T2DM at the PHCs in Yogyakarta City and are in charge in PROLANIS. Exclusion criteria are on leave from work; and 5) For BPJS Health Officers: a staff who manages and is responsible for the PROLANIS program within the scope of BPJS Health. Exclusion criteria are: officers who are on leave from work and officers who are conducting tasks out of town.

Phase 2 (Qualitative study). For the HE method, participants are experts who have experience in the field of computer science or digital health and understand the heuristic principles. For the TA method, participants who will be involved are patients diagnosed with T2DM for at least 1 year, aged 21 years old or older and are members of PROLANIS, family caregiver members who live at home. For the nurse: nurses who carry out the PROLANIS program and has a smartphone and can operate android-based applications.

Phase 3, (Quantitative study). The participants to be recruited are: 1) Patients with T2DM: willing to be participant, member of PROLANIS, aged 21 years old or older, diagnosed with T2DM for at least 1 year, has a smartphone and can operate it. Exclusion criteria are: patients with T1DM or participant who have history of diabetes since childhood, patients with T2DM who are pregnant, have comorbidities with CKD, or have mental disorders; 2) For family:

family caregiver aged 18 years old and over, family members who live in the same house as the patient, have a smartphone and can operate it. Exclusion criteria are: family members who work outside the home full day, with mental disorders or who are sick; 3) For nurses: a nurse who carry out the PROLANIS program at PHC where they are on duty and has a smartphone and can operate it. Exclusion criteria are: nurses who are on leave from work. Withdrawal criteria are: Patients with T2DM who withdrew, who had to be hospitalized, did not complete self-management results in first, second and third months, or did not follow HbA1C measurements before and after the intervention.

## Data analysis

Data analysis aims to determine the outcome of each phase of this study. For Phase 1 (Qualitative study), all data from FGD recordings will be transcribed verbatim and managed by using NVivo 12 software. The content analysis will start with the data encoding form of the verbatim transcripts from the FGDs (3 units of analysis) and in-depth interviews (12 units of analysis). We will read each transcript several times to determine the coding. Then, the codes that have the same meaning will be grouped into categories, which are then grouped into themes. The researchers will integrate each theme into a qualitative description. Codes and themes will be agreed upon by the researcher teams [31].

For Phase 2 (Qualitative study) in the HE method, data analysis at this phase will be done by grouping the problems found by the experts according to the HE principle. The list of problems that have been identified and agreed upon by the experts is given an assessment of the severity of the problem and calculated as the absolute severity of each problem and reported according to the severity rating table in Table 1 [23].

In the TA method, the problems found can arise from the comments and behaviour of the participants recorded when using the IDSM Application. The audio recording data will be transcribed verbatim by the researchers and cross-checked for accuracy of representation among the researcher team. The results of the transcript will be analysed to find problems in the operation of the application software carried out by users. The results of this analysis will be in the form of a group of problems found in the operation of the application software. The data will be processed using descriptive statistics. Furthermore, a rating will be given to each assessment parameter that is analysed [32].

Phase 3 (Quantitative study), hypothesis testing will be conducted in stages: univariate analysis, bivariate and multivariate analyses. First, the data normality test will be done using Kolmogorov Smirnov tests. Univariate analysis aims to explain the characteristics of each variable studied. Mean/median will be used to describe the participant's characteristics based on age and duration of DM. Participant's gender, education and profession will be presented using distribution of frequencies and percentages. Dependent t-test will be used to analyse the difference in mean of self-management scores before and after using the IDSM App in each group if

**Table 1. Severity rating.**

| Scale | Term | Description |
|---|---|---|
| 0 | No problem | Not a usability problem. |
| 1 | Cosmetic problem | The problem is only on the display side, it has no effect on user comfort. |
| 2 | Minor Problem | Problems need improvement but with little priority. |
| 3 | Major Problem | Problems need high priority fix. |
| 4 | Catastrophe | The problems received by users are very large, complex and harm the users, improvements must be made. |

the data is normal distributed. Otherwise, Wilcoxon test will be used. Summary Statistical Measure will be used to analyse the difference in mean of self-management scores between the experiment group and the control group after intervention if the data are normal distributed. Otherwise, Mann-Whitney test will be used. Dependent t-test will be used to analyse the difference in mean of HbA1c value before and after intervention, if the data is normal distributed and if not Wilcoxon test will be used as alternative tests. Independent t-test will be used to analyse the difference mean of HbA1c value between the experiment group and the control group after intervention if the data is normal distributed and Mann-Whitney test will be used as an alternative test. Multiple Regression will be used to analyse the most dominant variables affecting self-management and glycemic control. Size effect of the intervention will be measured using Cohen's formula with the category of strength effectiveness according to Cohen is 0.2 if the effectiveness is less, 0.5 if the effectiveness is moderate and 0.8 if the effectiveness is strong [33]. Data processing will be conducted based on the data collected using the SPSS 22 program (IBM Corp., Armonk, NY) which is subscribed to by Universitas Gadjah Mada.

## Security and data management plans

All data collected during the study will be kept confidential and all researcher team members will be bound by data confidentiality. The data collected from all phases will be anonymized and stored securely in a computer database. Quantitative data will be anonymized for statistical analysis. The analysis of all qualitative data will be conducted anonymously in pseudonymous form (i.e., without the name of the person, institution or location). The recorded data will be deleted after the study is completed if there is a request from the participants. For reporting the results of the qualitative study, the Consolidated Criteria for Reporting Qualitative Research (COREQ) checklist will be used [34]. Reporting the results of the quantitative study will use the Transparent Reporting of Evaluations with Nonrandomized Designs (TREND) checklist [35]. This study will provide opportunities for other researchers who need those data by contacting the corresponding author. Dissemination of findings and data sharing will be done after this research is completed. The data will be provided as part of the manuscript. Once we finished with data collection.

## Expected results

Phase 1 (Qualitative study). The outcome of this study will be described through the themes found which will later be used as the content of the application software that will be developed. Phase 2 (Qualitative study). Based on the results of phase 1 and the literature review, we will develop an android-based IDSM Application that has been usability tested and can be readily applied. Phase 3 (Quantitative study), the main outcomes expected in this phase are the android-based IDSM Application that effectively improves self-management and glycemic control of Patients with T2DM.

## Ethical considerations

We will use an information sheet and a consent form as prove that all participants are well informed and agree to participate. All participants will be given the information verbally in addition to written information in detail about the purpose, benefits and possible side effects, confidentiality of information, freedom to refuse and can resign at any time during the study. For participants who cannot read or write, an explanation about the research and informed consent will be read for them by researcher and then participants are asked to give a thumbprint as proof of their willingness to take part in the study. Patients who can't buy or will not have smartphone will be given a module after research finish. Each participant has the right to

refuse or discontinue participation at any time before or during data collection, without giving reasons. Ethical approval for conducting this research was obtained from the Medical and Health Research Ethics Committee of the Faculty of Medicine, Public Health and Nursing Universitas Gadjah Mada with the number of: KE/FK/1068/EC/2021 on September 27, 2021.

## Timeline of the study

The estimated time for this study is ten months which is divided into three phases. Phase 1 will be conducted for three months, phase 2 will be conducted for four months and phase 3 about three months.

## Study risks

The risk that may occur in this study is that not all Patients with T2DM who are members of PROLANIS in Yogyakarta city use android-based applications in carrying out their self-management, so it is possible that many patients still cannot operate the applications. To anticipate these problems, researchers will provide training on how to use the IDSM application and will provide assistance until participants can operate the application software. In addition, to anticipate errors in entering self-managed data for three months, the researchers will monitor every day through the data server and provide reminders to the participants.

## Strengths and limitation

This study offers availability of an integrated diabetes self-management program using Indonesia-language. An android application that involves patients, families and nurses and is useful to guide patients and facilitate monitoring of families and nurses to pay more attention to the implementation of the self-management of patients with diabetes in order to improve their self-management and glycemic control. This android-based IDSM Application can also be used as a means to evaluate the implementation of PROLANIS program at the PHC level, especially among Patients with T2DM.

One limitation in this study is that this research specially in phase 3 will be conducted in three months which might have the potential of participants to drop out before the study is finish. To anticipate this, researchers will monitor and evaluate every day through the server and conduct self-management measurements once a month.

## Dissemination and implementation

This study will provide opportunities for other researchers who need our data by contacting the corresponding author. Dissemination of findings and data sharing will be done after this research is completed. To facilitate data access and data protection during the research process, we will report any research progress to all teams regularly. This study's results will be published in reputable international journals. The results of this study will be applied in the development of medical surgical nursing, especially in the management of diabetes. The results of this study will also provide input for the government, especially BPJS Health, that this android-based IDSM Application can be used as a means or media in supporting the implementation of self-management of patients with T2DM who are PROLANIS members in Indonesia.

## Conclusion

This study protocol explains the purpose, significance and scope of the mixed methods approach as a comprehensive research design. The findings of this study are expected to help optimize the implementation of diabetes self-management so that it can improve glycemic

control of patients with T2DM as one of the indicators of the PROLANIS program at PHCs in Yogyakarta city. The aim of the research team to publish this protocol is to increase the transparency of the study and can provide a reference for other researchers so that the published research protocols can prevent unnecessary duplication.

## Supporting information

**S1 File. FGD guideline for patient with type 2 diabetes mellitus.**
(PDF)

**S2 File. FGD guideline for family.**
(PDF)

**S3 File. FGD guideline for nurse.**
(PDF)

**S4 File. Interview guideline for the doctor.**
(PDF)

**S5 File. Interview guideline for staff in charge of BPJS Health.**
(PDF)

**S6 File. Heuristic evaluation guideline.**
(PDF)

**S7 File. Think aloud guideline.**
(PDF)

**S8 File. IDN-DSMI 35 questionnaire.**
(PDF)

## Author Contributions

**Conceptualization:** Dewi Murdiyanti Prihatin Putri, Yoyo Suhoyo, Ariani Arista Putri Pertiwi, Christantie Effendy.

**Data curation:** Dewi Murdiyanti Prihatin Putri.

**Formal analysis:** Dewi Murdiyanti Prihatin Putri.

**Funding acquisition:** Dewi Murdiyanti Prihatin Putri.

**Investigation:** Dewi Murdiyanti Prihatin Putri.

**Methodology:** Dewi Murdiyanti Prihatin Putri, Yoyo Suhoyo, Ariani Arista Putri Pertiwi, Christantie Effendy.

**Project administration:** Dewi Murdiyanti Prihatin Putri, Christantie Effendy.

**Resources:** Dewi Murdiyanti Prihatin Putri.

**Software:** Dewi Murdiyanti Prihatin Putri.

**Supervision:** Dewi Murdiyanti Prihatin Putri, Yoyo Suhoyo, Ariani Arista Putri Pertiwi, Christantie Effendy.

**Validation:** Dewi Murdiyanti Prihatin Putri, Yoyo Suhoyo, Ariani Arista Putri Pertiwi, Christantie Effendy.

**Visualization:** Dewi Murdiyanti Prihatin Putri, Christantie Effendy.

Writing – **original draft:** Dewi Murdiyanti Prihatin Putri.

Writing – **review & editing:** Dewi Murdiyanti Prihatin Putri, Yoyo Suhoyo, Ariani Arista Putri Pertiwi, Christantie Effendy.

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
