## [Decision Letter · Decision Letter 0]

8 Jun 2022

PONE-D-21-37892Effectiveness of Android-based Integrated Diabetes Self-Management (IDSM) Application improving self-management and glycemic control of type 2 diabetes mellitus patients in Indonesia: a mixed methods study protocolPLOS ONE

Dear Dr. Effendy,

Thank you for submitting your manuscript to PLOS ONE. After careful consideration, we feel that it has merit but does not fully meet PLOS ONE’s publication criteria as it currently stands. Therefore, we invite you to submit a revised version of the manuscript that addresses the points raised during the review process.

The manuscript has been evaluated by two reviewers, and their comments are available below.

The reviewers have raised a number of concerns that need attention. They request additional information on methodological aspects of the study, including details regarding planned participants and recruitment, and they request further elaboration on the rationale for the study given the existence of similar studies in the literature. Please note that while the reviewer reference novelty PLOS ONE does not consider the novelty/impact of a study as part of our publication criteria; however we do require that the manuscript describes original research which is well justified and described, and analyses are performed to a high technical standard and are described in sufficient detail. The reviewers also requests copyediting of the manuscript to improve language and clarity

Could you please revise the manuscript to carefully address the concerns raised?

We look forward to receiving your revised manuscript.

Kind regards,

Jamie Royle

Staff Editor

PLOS ONE

Journal Requirements:

The funders had and will not have a role in study design, data collection and analysis, decision to publish, or preparation of the manuscript.

4. Please upload a copy of Supporting Information Figure 9 and 10 which you refer to in your text on page 18.

Reviewers' comments:

Reviewer's Responses to Questions

**Comments to the Author**

1. Does the manuscript provide a valid rationale for the proposed study, with clearly identified and justified research questions?

Reviewer #1: Partly

Reviewer #2: Yes

2. Is the protocol technically sound and planned in a manner that will lead to a meaningful outcome and allow testing the stated hypotheses?

Reviewer #1: Partly

Reviewer #2: Yes

3. Is the methodology feasible and described in sufficient detail to allow the work to be replicable?

Reviewer #1: No

Reviewer #2: No

4. Have the authors described where all data underlying the findings will be made available when the study is complete?

Reviewer #1: No

Reviewer #2: No

5. Is the manuscript presented in an intelligible fashion and written in standard English?

Reviewer #1: No

Reviewer #2: No

6. Review Comments to the Author

You may also provide optional suggestions and comments to authors that they might find helpful in planning their study.

Reviewer #1: The protocol entitled “Effectiveness of Android-based Integrated Diabetes Self-Management (IDSM) Application improving self-management and glycemic control of type 2 diabetes mellitus patients in indonesia: a mixed methods study protocol” expected to help optimize the implementation of diabetes self-management which can improve glycemic control of T2DM patients as one of the indicators of the Chronic Disease Care Program [Program Pengelolaan Penyakit Kronis (PROLANIS)] program in the PHC setting. The authors emphasized that there is no research publication that develop an Android-based integrated diabetes self-management application in Indonesia. However, many studies those application-based diabetes control in different countries of the World are available in the literature. The novelty of this study is not well described. The organizing of the manuscript is little complicated. The information periodically turns to the data analyze. The writing style must move away from a thesis style. I suggest to the authors to reorganize and summarize some parts for better understanding. Also, I believe there is no need to make a data analyze table.

Reviewer #2: 1. Use the same format of referencing in whole paper. There are (2) and [3].

2. There is no definition about diabetes self-management.

3. As you have a separate section called “study aims”, it is better to remove the last two line of “Background” that you wrote the aim of the study.

4. Will all the study phases be conducted in Public Health Center (PHC) in Yogyakarta city? How about for family members? How you will reach them?

5. What are the reason to do FGDs among patients in Phase 1? Will it let patients to participate in discussion by considering their own unique experiences? How about their gender, religious …composition?

6. Will other types of diabetes like T1DM… be considered as exclusion criteria?

7. How about a family member who has T2DM? will he/she be excluded?

8. What is sampling method for doctors?

9. Except “Peer debriefing and member checking” what other things you will do to increase trustworthiness? What about credibility? Transferability,..

10. How and why you will choose “two PHCs” for data collection in phase 3? Why not more?

11. There is no any information about data collection tool for qualitative phases? How they will be developed and used? How about the translation? How long interviews will be conducted approximately? Where the interviews/FGDs will be conducted?

12. There is no more information about study procedure. How the target groups for different phases will be chosen? Who will be involved to recruit or conduct interviews?

13. How about ethical considerations like using information sheet? Consent form? How many versions you use? How about those who can’t read or write?

14. Please use reference to support “Measurement of self-management scores will be done three times each month after the intervention..”

15. One of the questions is that considering only T2DM patients who use Smartphone and can operate it will make ethical issues? What about those patients who can’t buy or will not have smartphone?

16. This paper should be edited by an English native. For example, “Phase 2, phase 2b will use the Think Aloud (TA) method. In this phase the researcher tested the Android-based IDSM Application to the user t to find problems in operating it. Participants who will involved are participants who have been given an explanation and give a consent.”

7. PLOS authors have the option to publish the peer review history of their article (what does this mean?). If published, this will include your full peer review and any attached files.

Reviewer #1: No

Reviewer #2: **Yes: **Masoud Mohammadnezhad

---

## [Author Response · Author response to Decision Letter 0]

2 Aug 2022

We thank you very much for thoughtful feedback and the opportunity to revise our paper.

The comments have been very useful and helped to strengthen the paper. We have revised our manuscript accordingly. We included the reviewer’s comments and responded to them individually. We have included the page number to the checklist in the point by point response below. The purpose of this is to show how we addressed each response and to describe the changes that we have made. We have also carefully revised the language and flow of the text and proofread the paper. (Certificate of the proof read attached). 

This revision has been approved by all authors. We attached the revised manuscript and we uploaded it.

---

## [Decision Letter · Decision Letter 1]

22 Aug 2022

PONE-D-21-37892R1Integrated Diabetes Self-Management (IDSM) mobile application to improve self-management and glycemic control among patients with Type 2 Diabetes Mellitus (T2DM) in indonesia: a mixed methods study protocol.PLOS ONE

Dear Dr. Christantie Effendy,

Thank you for submitting your revised manuscript. After careful consideration, we feel that it has merit but does not fully meet PLOS ONE’s publication criteria as it currently stands. Therefore, we invite you to submit a revised version of the manuscript that addresses the points raised during the review process.

We look forward to receiving your revised manuscript.

Kind regards,

Jahanpour Alipour, Ph.D.

Academic Editor

PLOS ONE

Journal Requirements:

Additional Editor Comments:

The authors improved the manuscript. However, it is necessary to consider the following concerns;

-Authors should calculate the sample size considering the "two sided" test. The population is relatively small.

-The authors mentioned that the control group will receive standard treatment via WhatsApp chat. Is this a standard treatment in Indonesia? How frequently the nurses will send messages to the participants? Is that a group based chat or personal communication will be allowed? The treatment for control group should be given in detail.

-The age range of inclusion criteria should be reconsidered. How do you discriminate a Type 2 and Type 1 or Moody Type DM in a participant aged 18 years?

Reviewers' comments:

Reviewer's Responses to Questions

**Comments to the Author**

1. Does the manuscript provide a valid rationale for the proposed study, with clearly identified and justified research questions?

Reviewer #1: Partly

Reviewer #2: Yes

2. Is the protocol technically sound and planned in a manner that will lead to a meaningful outcome and allow testing the stated hypotheses?

Reviewer #1: Partly

Reviewer #2: Yes

3. Is the methodology feasible and described in sufficient detail to allow the work to be replicable?

Reviewer #1: Yes

Reviewer #2: Yes

4. Have the authors described where all data underlying the findings will be made available when the study is complete?

Reviewer #1: No

Reviewer #2: Yes

5. Is the manuscript presented in an intelligible fashion and written in standard English?

Reviewer #1: Yes

Reviewer #2: Yes

6. Review Comments to the Author

You may also provide optional suggestions and comments to authors that they might find helpful in planning their study.

Reviewer #1: The authors improved the manuscript. However, the following revisions are needed for better explaining;

-Authors should calculate the sample size considering the "two sided" test. The population is relatively small.

-The authors mentioned that the control group will receive standard treatment via Whatsapp chat. Is this a standard treatment in Indenosia? How frequently the nurses will send messages to the participants? Is that a group based chat or personal comminucation will be allowed? The treatment for control group should be given in detail.

-The age range of inclusion criteria should be reconsidered. How do you discriminate a Type 2 and Type 1 or Moody Type DM in a participant aged 18 years?

Reviewer #2: Thank you for addressing my comments. This study now sound very clear and can be published as it is.

7. PLOS authors have the option to publish the peer review history of their article (what does this mean?). If published, this will include your full peer review and any attached files.

Reviewer #1: No

Reviewer #2: **Yes: **Masoud Mohammadnezhad

---

## [Author Response · Author response to Decision Letter 1]

30 Sep 2022

• Sample size have calculate considering the “two sided” test, and it results 35 participants. To anticipate the possibility of participants dropping out, a number of samples will be added to keep them fulfilled by the formula : n’ = n/(1-f) so the sample to be used is 39 participants for each group (intervention and control group). (page 9 line 221).

• Regarding the use of whatsapp chat, patients may have personal communication with health provider at PHC in addition to group communication. They communicate via WA chat as needed. Whatsapp is very popular among Indonesian, most of Indonesian people who has smartphone using whatsapp including patients with diabetes. So it is easy for a health provider to communicate with all patients who are in Prolanis program using whatsapp, eventhough it is not written national standar.

The control group will use the whatsapp group to communicate or consult directly with a health provider. Self-management will be carried out independently by the patient in his or her home. Monitoring by a health provider will be carried out using whatsapp group and when patient come to the PHC for routine check up. (page 9 line 214 – 218).

• The age range for the inclusion criteria is participants aged 21 years old and older who have no history of diabetes since childhood. (page 11 line 254 , 256-257).

---

## [Decision Letter · Decision Letter 2]

21 Oct 2022

Integrated Diabetes Self-Management (IDSM) mobile application to improve self-management and glycemic control among patients with Type 2 Diabetes Mellitus (T2DM) in indonesia: a mixed methods study protocol.

PONE-D-21-37892R2

Dear Dr. Christantie Effendy,

We’re pleased to inform you that your manuscript has been judged scientifically suitable for publication and will be formally accepted for publication once it meets all outstanding technical requirements.

Kind regards,

Jahanpour Alipour, Ph.D.

Academic Editor

PLOS ONE

Reviewers' comments:

Reviewer's Responses to Questions

**Comments to the Author**

1. Does the manuscript provide a valid rationale for the proposed study, with clearly identified and justified research questions?

Reviewer #1: Yes

Reviewer #2: Yes

2. Is the protocol technically sound and planned in a manner that will lead to a meaningful outcome and allow testing the stated hypotheses?

Reviewer #1: Yes

Reviewer #2: Yes

3. Is the methodology feasible and described in sufficient detail to allow the work to be replicable?

Reviewer #1: Yes

Reviewer #2: Yes

4. Have the authors described where all data underlying the findings will be made available when the study is complete?

Reviewer #1: Yes

Reviewer #2: Yes

5. Is the manuscript presented in an intelligible fashion and written in standard English?

Reviewer #1: Yes

Reviewer #2: Yes

6. Review Comments to the Author

You may also provide optional suggestions and comments to authors that they might find helpful in planning their study.

Reviewer #1: The study titled "Integrated Diabetes Self-Management (IDSM) mobile application to improve self-management and glycemic control among patients with Type 2 Diabetes Mellitus (T2DM) in indonesia: a mixed methods study protocol" is accepted as it is.

Reviewer #2: Thank you for addressing the comments. I am happy with this version and recommend this paper to be published.

7. PLOS authors have the option to publish the peer review history of their article (what does this mean?). If published, this will include your full peer review and any attached files.

Reviewer #1: No

Reviewer #2: **Yes: **Masoud Mohammadnezhad

---

## [Editor Report · Acceptance letter]

2 Nov 2022

PONE-D-21-37892R2 

*Integrated Diabetes Self-Management (IDSM) mobile application to improve selfmanagement and glycemic control among patients with Type 2 Diabetes Mellitus (T2DM) in Indonesia: a mixed methods study protocol*

Dear Dr. Effendy:

I'm pleased to inform you that your manuscript has been deemed suitable for publication in PLOS ONE. Congratulations! Your manuscript is now with our production department. 

Kind regards, 

on behalf of

Dr., Jahanpour Alipour 

Academic Editor

PLOS ONE